# Socioeconomic inequalities in health among elderly people in Taiwan: A life course perspective

Hsiao-Hsiao Tan[ID], Yi-Chan Lee, Ya-Mei Chen, Tung-liang Chiang*

Institute of Health Policy and Management, College of Public Health, National Taiwan University, Taipei, Taiwan

* tlchiang@ntu.edu.tw

## Abstract

### Background

Individuals' health could be well predicted through their lifetime trajectories. Those with greater socioeconomic advantages tend to exhibit better health and a slower rate of health decline as they age, and vice versa.

### Objective

This study aims to investigate how health trajectory in old age is shaped by the accumulation of socioeconomic status (SES) across three life stages.

### Method

The dataset used in this study is derived from the Taiwan Longitudinal Study on Aging (TLSA) conducted from 1989 to 2003, with 4,048 respondents aged 60 and older. A stratified analysis, utilizing Generalized Estimating Equations (GEE) models, is employed to investigate the interrelationships between SES across three life stages and the longitudinal patterns of three health outcomes.

### Result

Of all the participants over the five waves, participants' health declined with aging, while the proportion of higher SES among remaining individuals tended to increase. A gradient in health improvements is observed, corresponding to the accumulation of SES, even after adjusting for demographic factors and baseline health. Moreover, improvements in a single SES indicator are significantly associated with better health when controlling for the other two indicators (all $p < 0.01$). This indicates an independent effect of each SES indicator. Additionally, SES tends to be inherited from preceding statuses. The interaction among all three SES indicators cannot be overlooked when considering the association between SES and health.

**Data availability statement:** The data supporting the findings of this study are accessible through the Survey Research Data Archive (SRDA) established by the National Academy of Taiwan. It is important to note that access to these data is restricted, as they were utilized under license for this study and are not publicly available. Interested researchers may obtain access by submitting an application to the SRDA (https://srda.sinica.edu.tw/). Additionally, data can be made available by the researchers upon reasonable request; inquiries should be directed to SRDA via email (srda@gate.sinica.edu.tw). Health Promotion Administration, Ministry of Health and Welfare (2015). 1989 Taiwan Longitudinal Study on Aging (AD040001) [data file]. Available from Survey Research Data Archive, Academia Sinica. https://doi.org/10.6141/TW-SRDA-AD040001-1 Health Promotion Administration, Ministry of Health and Welfare (2020). 1993 Taiwan Longitudinal Study on Aging (AD040002) [data file]. Available from Survey Research Data Archive, Academia Sinica. https://doi.org/10.6141/TW-SRDA-AD040002-1 Health Promotion Administration, Ministry of Health and Welfare (2020). 1996 Taiwan Longitudinal Study on Aging (AD040003) [data file]. Available from Survey Research Data Archive, Academia Sinica. https://doi.org/10.6141/TW-SRDA-AD040003-1 Health Promotion Administration, Ministry of Health and Welfare (2020). 1999 Taiwan Longitudinal Study on Aging (AD040004) [data file]. Available from Survey Research Data Archive, Academia Sinica. https://doi.org/10.6141/TW-SRDA-AD040004-1 Health Promotion Administration, Ministry of Health and Welfare (2020). 2003 Taiwan Longitudinal Study on Aging (AD040005) [data file]. Available from Survey Research Data Archive, Academia Sinica. https://doi.org/10.6141/TW-SRDA-AD040005-1

**Funding:** The author(s) received no specific funding for this work.

**Competing interests:** No authors have competing interests.

## Conclusion

Health outcomes are shaped progressively and independently by each SES factor, with these effects being reinforced by the cumulative nature of SES. Policies aimed at enhancing later-life health are better implemented in early life stages. However, it is never too late, as each SES at different life stages has its own effect and cannot be offset.

## Introduction

People's health can be well predicted through their lifetime trajectories. Individuals who enjoy greater socioeconomic advantages have better health and a slower rate of decline as they age, and vice versa [1,2]. Prior research has shown that childhood socioeconomic status independently influences adult health [3–5]. A life course perspective on chronic disease epidemiology illustrates the influence of early- and later-life biological, behavioral, social, and psychological exposures on the development of disease in a person [6,7].

The majority of previous studies described health at a single point in time. However, as life is a dynamic process, the relationship between socioeconomic status (SES) and health should not be viewed as singular events but as a continuum over time [8]. No single factor at any given time has a considerable effect on the health of an individual, as poor adult health results from the accumulation of numerous factors. Cumulative effects explain why continuing adverse situations exacerbate downward impacts [9]. Moreover, socioeconomic circumstances in childhood and adulthood are linked, and it has been difficult to isolate the specific effects of socioeconomic conditions at every life stage [10,11]. These temporal processes are interconnected and manifest themselves in disease trends [6].

As health and socioeconomic status are continuously intertwined throughout life, it is difficult to disentangle the causal processes involved [12]. Furthermore, a life course approach to aging research is more complicated because markers may not remain constant over secular time. Aging is linked to multi-system decline via common etiological mechanisms. In light of the stochastic nature of aging, this integrated life course model is unlikely to predict individual outcomes with great certainty, but it might be able to explain group differences in aging [13]. The purpose of this study is to investigate how health trajectory in old age is shaped by the accumulation of socioeconomic status across three life stages.

## Method

### Study data and design

The dataset utilized in this study is derived from five waves of the Taiwan Longitudinal Study on Aging (TLSA), conducted in 1989, 1993, 1996, 1999, and 2003. TLSA began as a cross-sectional study in 1989 and later continued to observe the same population in subsequent years, thus transitioning into a longitudinal study. Despite the inclusion of additional individuals in later waves, no new participants were included in this specific study.

The Taiwan Longitudinal Study on Aging (TLSA) is designed to comprehensively understand the social and economic conditions, along with the health and well-being, of older individuals. It assesses characteristics across multiple domains to provide high-quality multidisciplinary data. These domains include: (1) background information, (2) family composition and social support network, (3) health status and disease history, (4) work history, (5) social behaviors and attitudes, (6) residence history, (7) financial status, and (8) mental status from a social capital perspective.

The 1989 survey was conducted by the Taiwan Provincial Institute of Family Planning in collaboration with the Population Studies Center at the University of Michigan. A stratified three-stage systematic random sampling method was employed to select 4,412 participants aged 60 years and above, with the sampling frame based on household registration data. The study population, as of late 1988, resided in 361 administrative units across Taiwan, excluding 30 mountainous areas predominantly inhabited by indigenous populations. The sampling process was conducted in three stages: first, 56 of 331 townships were selected; second, blocks within these townships were sampled; and finally, two elderly individuals (aged ≥60) from each selected block were chosen as survey subjects. With a population of approximately 1.81 million individuals aged 60 and over from 331 non-mountainous townships and districts, the sampling probability was determined to be 1/410. The age and sex distributions of the sample were then compared to the official household registration data from the end of 1987. In general, the survey results demonstrate representativeness [14].

In total, 4,049 individuals completed the baseline survey [14]. In-person interviews were conducted throughout all five waves. The baseline interviews in 1989 survey achieved a response rate of 92%, while subsequent follow-up surveys maintained an approximately 90% response rate among continuing participants. Of the initial 4,049 participants, 1,743 were retained in the study for the 2003 survey, while 2,306 were lost to follow-up. Detailed descriptions of this study presented in previous publications [15–17].

A copy of the dataset was subsequently transferred to and archived in the Survey Research Data Archive of Academia Sinica. Therefore, this study utilized the secondary analysis of de-identified data. The study was approved and classified as exempt by the Research Ethics Committee of National Taiwan University (NTU), with the approval number/reference: 202309HS009. It should be noted that due to an age recording error for one participant, the final sample for this study comprised 4,048 respondents. The sample for each wave, alongside the response and attrition rates, is presented in the flowchart in Fig 1.

This study aims to explore how cumulative exposure to socioeconomic status (SES) at different life stages shapes health trajectories in later life. SES data were collected in the baseline year, and health trajectories were analyzed using Generalized Estimating Equations (GEE). This design allowed for the examination of baseline health and subsequent changes over time.

### Variables

**Socioeconomic status (SES) indicators.** This study utilized socioeconomic status (SES) indicators across three life stages: childhood, early adulthood, and late adulthood. These indicators included the educational attainment of participants' fathers, the participants' own educational attainment, and monthly household income. All variables were measured at the baseline year (1989).

The educational attainment of respondents' fathers was categorized into two groups based on their literacy. On the other hand, to enhance comparability, early and late adulthood SES indicators were divided into three groups rather than a dichotomous classification. Participants' education was measured based on the highest level of education achieved and categorized into three groups: illiterate, primary education, and more than primary. Respondents' monthly household income levels were categorized into three strata: low (< 3,000 NTD), medium (3,000–19,999 NTD), and high (≥ 20,000 NTD). According to Taiwan's Ministry of Health and Welfare, the minimum living wage in 1991 was 3,000 NTD. The Directorate-General of Budget, Accounting and Statistics, Executive Yuan, reported an average salary of 21,247 NTD in 1989. Therefore, we used 3,000 NTD and 20,000 NTD as cut points to divide participants into three

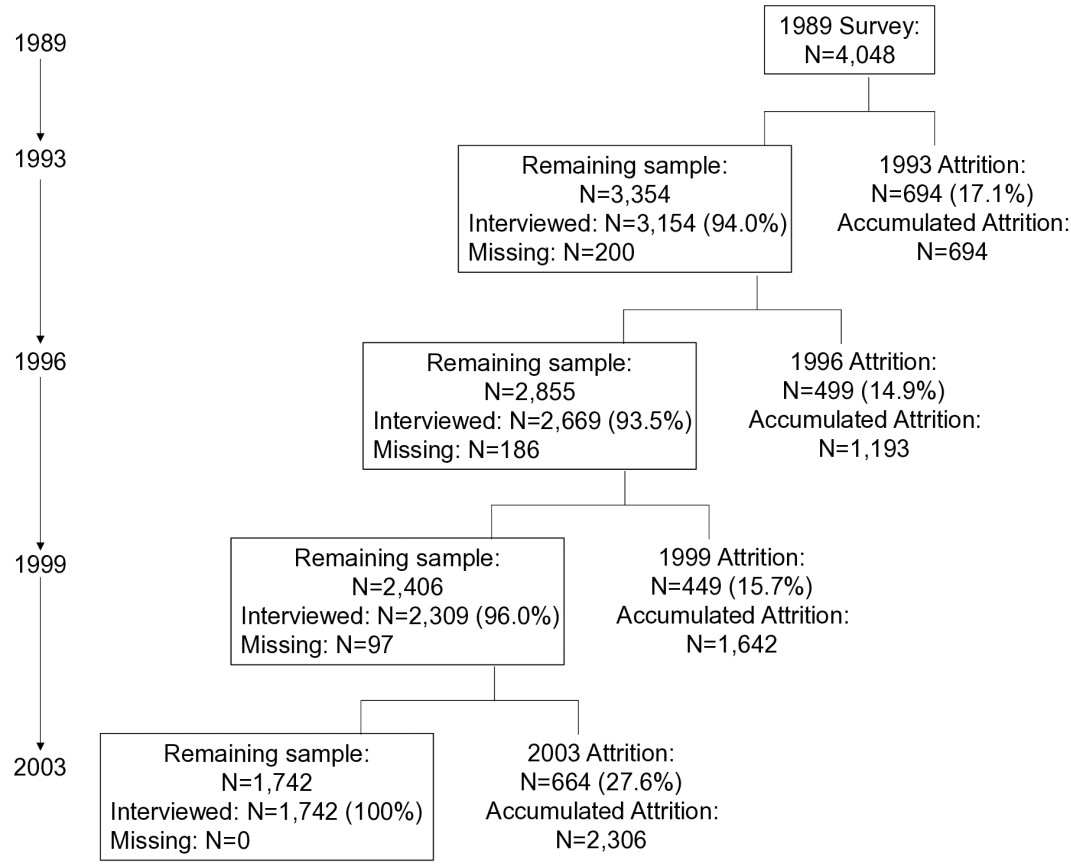

**Fig 1. Participants in the TLSA from 1989 to 2003.**

income groups. To address nonresponse in the monthly household income data, hot-deck imputation was employed using the modal category by sex and age, as this method is recognized for its computational simplicity and effectiveness [18,19].

**Health outcomes.** The health outcomes were measured using three indicators: self-rated health (SRH), activities of daily living (ADL) [20] disability (bathing only) and instrumental activities of daily living (IADL) [21] disability. The health outcomes were categorized as either "good" or "poor" based on self-rated health (SRH), and as "functionally independent" or "limited" in terms of ADL and IADL. Aside from general self-described health status, functional limitations were included with the aim of understanding disability trends among the study population. Due to the incomplete ADL data in the 1989 interview, only bathing was selected as the representative ADL indicator for this study. In Taiwan, difficulty bathing is the most frequently cited limitation in ADL. Having persistent difficulties bathing is an independent risk of long-term nursing home admission [22]. Moreover, due to significant missing data in the ADL for the year 1996, bathing-related data for that year were excluded from the analysis.

Notably, individuals categorized as "lost to follow-up" encompassed those who went missing and those who passed away. Unfortunately, this study could not distinguish between these two groups. Based on mortality data from the 2007 TLSA research report, the calculated mortality rate among individuals classified as "lost to follow-up" across the five study waves ranged from 83.9% to 92.5%. Hence, for analytical purposes, we categorized those classified as "lost to follow-up" as part of the unhealthy group.

## Statistical analyses

Generalized Estimating Equations (GEE) models, recognized for their ability to handle repeated measures and specify flexible correlation structures, were employed in this study. Logistic regression within the framework of GEE was applied to analyze longitudinal patterns of three health outcomes, with indicators aggregated into two groups and reported for each wave.

Utilizing three SES indicators, the sample was subdivided into 18 discrete groups. The subgroup exhibiting the lowest SES was utilized as the reference group to assess comparative health improvements across the more privileged subgroups. A stratified GEE analysis was performed to assess the relationship between SES across three life stages and health outcomes [23].

Sensitivity analyses were conducted to investigate the health outcomes with varying proportions of individuals "lost to follow-up" classified as unhealthy. We randomly assigned 10%, 30%, 50%, 70%, 80% and 90% of individuals "lost to follow-up" as unhealthy, with the remainder classified as healthy. Additionally, we evaluated the outcomes when all lost to follow-up individuals were excluded.

## Result

The characteristics of participants, along with the number of follow-up losses across five waves of the TLSA, are presented in Table 1. Of the initial 4,048 participants, 42.9% were women, with an average baseline age of 67.8 years. A total of 48.1% of participants' fathers were illiterate, 39.8% were literate, and 12.1% lacked relevant data. Approximately 40% of participants were either illiterate or had only received primary education, with less than 20% attaining education more than primary. After implementing Hot-deck imputations, 18.9% of respondents were classified in the low household income group, 63.6% in the medium group, and 17.5% in the high-income group. The response rates for each wave exceeded 91.7%, with attrition rates ranging from 14.9% to 27.6%. By 2003, 1,742 participants remained in the study.

Over the five waves, participants' health declined with aging, while the proportion of higher SES among remaining participants tended to increase. Data regarding sociodemographic characteristics and baseline health, segregated by participants' educational attainment, are outlined in Table 2. Individuals with lower educational attainment are more inclined to be female, older, have an illiterate father, possess lower monthly household income, and exhibit poorer health conditions.

Tables 3–5 display the health outcomes obtained via GEE analysis. Generally, populations characterized by more advantageous SES tend to exhibit better overall health outcomes, even after controlling for demographic factors and baseline health. Individuals experiencing three SES disadvantages exhibit the poorest health, demonstrating a gradual improvement as the number of advantageous SES factors increases. The population with the most advantaged SES generally exhibits the best health outcomes. Compared to individuals with disadvantaged SES across all life stages, those with advantaged SES in all three stages have 3.06 times the odds (95% CI: 2.39–3.90) of reporting good SRH after adjusting for sex and age. For ADL, they have 1.58 times the odds (95% CI: 1.27–1.96) of reporting independence after adjustment. For IADL, they have 4.49 times the odds (95% CI: 3.53–5.72) of reporting independence after adjustment. Accumulation of advantaged and disadvantaged SES across three life stages sets the baseline health. More advantaged SES allows for a greater baseline health reserve to be diminished by aging. Following adjustments for gender, age and all baseline health conditions in Model 3 of Tables 3–5, the disparities in health outcomes among various SES subgroups have diminished greatly. This suggests a more substantial impact of SES indicators on baseline health compared to their influence on the trajectory of health over time. Yet, the rate of decline is influenced by SES accumulation, with increased SES accumulation significantly associated with a lesser decline in health. The participants' aging process differs across subgroups with diverse SES indicators.

Nevertheless, each SES characteristics contributes its distinct effect to health outcomes. Among the three SES indicators, improvements in a single SES indicator are associated with better health when controlling for the other two indicators. Moreover, despite having the same SES status at a given moment, the health of subgroups varies because of their

**Table 1. Attributes of participants through five waves of the Taiwan Longitudinal Study on Aging (TLSA).**

| Year | 1989 | | 1993 | | 1996[a] | | 1999 | | 2003 | |
|---|---|---|---|---|---|---|---|---|---|---|
| | N | % | N | % | N | % | N | % | N | % |
| **Statement** | | | | | | | | | | |
| Total survival | 4048 | 100.0% | 3354 | 100.0% | 2855 | 100.0% | 2406 | 100.0% | 1742 | 100.0% |
| Interviewed | | | 3154 | 94.0% | 2669 | 93.5% | 2309 | 96.4% | 1742 | 100% |
| Missing | | | 200 | 6.0% | 186 | 6.5% | 97 | 4.0% | 0 | 0.0% |
| Loss follow up | | | 694 | | 1193 | | 1642 | | 2306 | |
| **Demographic characteristics** | | | | | | | | | | |
| *Gender* | | | | | | | | | | |
| Male | 2311 | 57.1% | 1894 | 56.5% | 1598 | 56.0% | 1325 | 55.1% | 924 | 53.0% |
| Female | 1737 | 42.9% | 1460 | 43.5% | 1257 | 44.0% | 1081 | 44.9% | 818 | 47.0% |
| *Age group (as of 1989)* | | | | | | | | | | |
| 60–64 years | 1562 | 38.6% | 1408 | 42.0% | 1286 | 45.0% | 1161 | 48.3% | 950 | 54.5% |
| 65–69 years | 1129 | 27.9% | 980 | 29.2% | 847 | 29.7% | 723 | 30.0% | 524 | 30.1% |
| 70–74 years | 710 | 17.5% | 564 | 16.8% | 459 | 16.1% | 344 | 14.3% | 199 | 11.4% |
| 75–79 years | 417 | 10.3% | 287 | 8.6% | 199 | 7.0% | 136 | 5.7% | 55 | 3.2% |
| 80 years and over | 230 | 5.7% | 115 | 3.4% | 64 | 2.2% | 42 | 1.7% | 14 | 0.8% |
| **Socioeconomic status (SES) indicators (surveyed at baseline year)** | | | | | | | | | | |
| *Father's education* | | | | | | | | | | |
| Illiterate | 1948 | 48.1% | 1628 | 48.5% | 1379 | 48.3% | 1156 | 48.0% | 810 | 46.5% |
| Literate | 1612 | 39.8% | 1383 | 41.2% | 1205 | 42.2% | 1026 | 42.6% | 771 | 44.3% |
| Missing | 488 | 12.1% | 343 | 10.2% | 271 | 9.5% | 224 | 9.3% | 161 | 9.2% |
| *Education* | | | | | | | | | | |
| Illiterate | 1675 | 41.4% | 1333 | 39.7% | 1104 | 38.7% | 909 | 37.8% | 625 | 35.9% |
| Primary education | 1595 | 39.4% | 1366 | 40.7% | 1167 | 40.9% | 990 | 41.1% | 725 | 41.6% |
| More than primary | 760 | 18.8% | 642 | 19.1% | 574 | 20.1% | 497 | 20.7% | 383 | 22.0% |
| Missing | 18 | 0.4% | 13 | 0.4% | 10 | 0.4% | 10 | 0.4% | 9 | 0.5% |
| *Monthly household income* | | | | | | | | | | |
| Low | 765 | 18.9% | 595 | 17.7% | 466 | 16.3% | 374 | 15.5% | 248 | 14.2% |
| Medium | 2575 | 63.6% | 2145 | 64.0% | 1838 | 64.4% | 1554 | 64.6% | 1119 | 64.2% |
| High | 708 | 17.5% | 614 | 18.3% | 551 | 19.3% | 478 | 19.9% | 375 | 21.5% |
| **Health indicators (surveyed at each wave)** | | | | | | | | | | |
| *Self-rated health* | | | | | | | | | | |
| Good | 1527 | 37.7% | 1251 | 37.3% | 765 | 26.8% | 626 | 26.0% | 467 | 26.8% |
| Poor | 2378 | 58.7% | 1717 | 51.2% | 1654 | 57.9% | 1683 | 70.0% | 1275 | 73.2% |
| Missing | 143 | 3.5% | 386 | 11.5% | 436 | 15.3% | 97 | 4.0% | 0 | 0.0% |
| *ADL disability (bathing only)* | | | | | | | | | | |
| Functionally independent | 3803 | 93.9% | 2951 | 88.0% | 333 | 11.7% | 2014 | 83.7% | 1442 | 82.8% |
| Limited | 241 | 6.0% | 198 | 5.9% | 221 | 7.7% | 295 | 12.3% | 300 | 17.2% |
| Missing | 4 | 0.1% | 205 | 6.1% | 2301 | 80.6% | 97 | 4.0% | 0 | 0.0% |
| *IADL disability* | | | | | | | | | | |
| Functionally independent | 2800 | 69.2% | 2237 | 66.7% | 1838 | 64.4% | 1482 | 61.6% | 979 | 56.2% |
| Limited | 1243 | 30.7% | 911 | 27.2% | 822 | 28.8% | 822 | 34.2% | 762 | 43.7% |
| Missing | 5 | 0.1% | 206 | 6.1% | 195 | 6.8% | 102 | 4.2% | 1 | 0.1% |

[a] The ADL data for 1996 were excluded from the analysis due to significant missing.

**Table 2. Characteristics of participants stratified by educational attainment in 1989.**

| Characteristics (N = 4048) | Participants' education | | | | |
|---|---|---|---|---|---|
| | Illiterate | Primary education | More than primary | Missing | P value |
| Gender (% of male) | 30.0% (N = 1675) | 71.9% (N = 1595) | 85.4% (N = 760) | N = 18 | 0.0001 |
| Age (median age) | 69.6 (N = 1675) | 66.7 (N = 1595) | 66.0 (N = 760) | N = 18 | 0.0001 |
| Father's education (% of literate) | 22.1% (N = 1426) | 49.0% (N = 1411) | 83.9% (N = 709) | N = 502 | 0.0001 |
| Monthly household income (% of high) | 7.4% (N = 1675) | 16.1% (N = 1595) | 42.2% (N = 760) | N = 18 | 0.0001 |
| Self-rated health (% of good) | 29.1% (N = 1582) | 41.2% (N = 1564) | 55.9% (N = 744) | N = 158 | 0.0001 |
| ADL disability (bathing only) (% of functionally independent) | 91.5% (N = 1674) | 95.5% (N = 1594) | 96.7% (N = 759) | N = 21 | 0.0001 |
| IADL disability (% of functionally independent) | 46.8% (N = 1675) | 82.6% (N = 1595) | 90.4% (N = 760) | N = 18 | 0.0001 |

**Table 3. GEE analysis of Self-rated health (SRH): Odds ratios (95% confidence intervals) across three life stages' SES strata.**

| Father's education | Education | Monthly household income | | |
|---|---|---|---|---|
| | | Low | Medium | High |
| **Model 1: crude model** | | | | |
| Illiterate | Illiterate | 1.00 | 1.43**(1.15–1.77) | 1.76**(1.17–2.67) |
| | Primary education | 1.46*(1.03–2.07) | 2.17***(1.75–2.70) | 2.88***(2.10–3.95) |
| | More than primary | 3.11**(1.51–6.42) | 3.12***(2.17–4.50) | 4.55***(2.80–7.39) |
| Literate | Illiterate | 1.35(0.94–1.95) | 1.55**(1.18–2.03) | 2.82**(1.46–5.46) |
| | Primary education | 1.50*(1.02–2.21) | 2.21***(1.78–2.75) | 3.22***(2.48–4.17) |
| | More than primary | 2.95***(1.61–5.41) | 3.17***(2.51–4.00) | 4.73***(3.73–5.98) |
| **Model 2: adjusted for gender and age** | | | | |
| Illiterate | Illiterate | 1.00 | 1.23(0.99–1.52) | 1.40(0.93–2.10) |
| | Primary education | 1.20(0.85–1.69) | 1.51***(1.20–1.89) | 1.89***(1.38–2.59) |
| | More than primary | 2.49**(1.24–4.97) | 2.10***(1.46–3.04) | 2.77***(1.70–4.50) |
| Literate | Illiterate | 1.43(0.98–2.08) | 1.40*(1.06–1.84) | 2.46**(1.30–4.68) |
| | Primary education | 1.26(0.86–1.84) | 1.60***(1.28–2.00) | 2.23***(1.70–2.91) |
| | More than primary | 2.14*(1.12–4.09) | 2.23***(1.76–2.84) | 3.06***(2.39–3.90) |
| **Model 3: adjusted for gender, age and baseline health** | | | | |
| Illiterate | Illiterate | 1.00 | 1.13(0.95–1.35) | 1.26(0.89–1.78) |
| | Primary education | 1.28(0.98–1.68) | 1.22*(1.02–1.47) | 1.69***(1.25–2.29) |
| | More than primary | 1.20(0.68–2.12) | 1.82***(1.28–2.58) | 1.62*(1.04–2.53) |
| Literate | Illiterate | 1.38(0.99–1.92) | 1.31*(1.03–1.66) | 2.28**(1.34–3.87) |
| | Primary education | 1.35(0.96–1.91) | 1.40***(1.16–1.69) | 1.58***(1.24–2.02) |
| | More than primary | 1.87*(1.06–3.28) | 1.68***(1.36–2.07) | 2.07***(1.68–2.57) |

* p < 0.05, ** p < 0.01, *** p < 0.001.

**Table 4. GEE analysis of ADL disability (bathing only): odds ratios (95% confidence intervals) across three life stages' SES strata.**

| Father's | Education | Monthly household income | | |
|---|---|---|---|---|
| education | | Low | Medium | High |
| **Model 1: crude model** | | | | |
| Illiterate | Illiterate | 1.00 | 1.44***(1.24–1.68) | 1.50*(1.08–2.07) |
| | Primary education | 1.13(0.87–1.45) | 1.60***(1.36–1.88) | 2.16***(1.58–2.94) |
| | More than primary | 1.70(0.69–4.18) | 1.72**(1.24–2.38) | 4.30***(2.22–8.34) |
| Literate | Illiterate | 1.18(0.88–1.58) | 1.40**(1.14–1.71) | 1.65(0.98–2.78) |
| | Primary education | 1.13(0.83–1.54) | 1.74***(1.47–2.05) | 2.48***(1.89–3.24) |
| | More than primary | 1.65(0.87–3.14) | 1.69***(1.39–2.05) | 2.40***(1.94–2.95) |
| **Model 2: adjusted for gender and age** | | | | |
| Illiterate | Illiterate | 1.00 | 1.17*(1.01–1.36) | 1.03(0.75–1.41) |
| | Primary education | 1.10(0.86–1.41) | 1.15(0.97–1.37) | 1.44*(1.06–1.96) |
| | More than primary | 1.65(0.73–3.69) | 1.28(0.92–1.77) | 2.60**(1.40–4.83) |
| Literate | Illiterate | 1.17(0.87–1.58) | 1.11(0.91–1.35) | 1.25(0.76–2.07) |
| | Primary education | 1.04(0.77–1.42) | 1.25**(1.06–1.49) | 1.65***(1.26–2.16) |
| | More than primary | 1.33(0.74–2.40) | 1.28*(1.05–1.56) | 1.58***(1.27–1.96) |
| **Model 3: adjusted for gender, age and baseline health** | | | | |
| Illiterate | Illiterate | 1.00 | 1.06(0.93–1.22) | 0.98(0.71–1.34) |
| | Primary education | 1.13(0.89–1.44) | 1.08(0.92–1.27) | 1.42*(1.05–1.92) |
| | More than primary | 1.70(0.75–3.83) | 1.28(0.94–1.75) | 2.26*(1.21–4.19) |
| Literate | Illiterate | 1.14(0.89–1.47) | 0.98(0.81–1.18) | 1.13(0.69–1.85) |
| | Primary education | 0.99(0.75–1.31) | 1.13(0.96–1.32) | 1.52**(1.16–1.98) |
| | More than primary | 1.25(0.69–2.28) | 1.16(0.96–1.40) | 1.41***(1.15–1.74) |

* $p < 0.05$, ** $p < 0.01$, *** $p < 0.001$.

differing life trajectories. For instance, individuals with the highest income yet lower educational attainment demonstrate worse health in comparison to those whose SES indicators are consistently advantaged.

Additionally, under the most disadvantaged circumstances in early and late adulthood SES, better childhood SES does not significantly contribute to health enhancement. The influence of a father's literacy on health outcomes is less than that of the other two indicators. This implies that father's educational attainment displays the least impact on health among all three SES indicators. Conversely, both early and late adulthood SES strongly associate with all three health outcomes.

Among SES indicators, educational attainment plays a more crucial role. Advancing educational attainment is more likely to result in greater health improvements compared to increases in monthly household income and father's educational attainment, particularly for IADL. Under constant childhood SES, participants who improve their educational attainment while maintaining household income have better health outcomes than those who increase household income while keeping educational levels unchanged.

The sensitivity analyses yield similar results to the study. Comparing the sample size of those lost to follow-up with mortality count data from the 2007 TLSA research report, the calculated mortality rate among individuals classified as "lost to follow-up" across five study waves ranged from 83.9% to 92.5%. The results for the 80% and 90% scenarios are more closely aligned with the study results. Although the magnitude attenuates, the overall associations and patterns remain stable. On the other hand, given that participants lost to follow-up were randomly classified as unhealthy while the others were assigned to the healthy group, the results for the 10% and 30% scenarios exhibit greater instability, with the 95% confidence intervals being comparably wider across all three health indicators (SRH, ADL, and IADL) generally. As the

**Table 5. GEE analysis of IADL disability: odds ratios (95% confidence intervals) across three life stages' SES strata.**

| Father's education | Education | Monthly household income | | |
|---|---|---|---|---|
| | | Low | Medium | High |
| **Model 1: crude model** | | | | |
| Illiterate | Illiterate | 1.00 | 2.05***(1.68–2.50) | 2.27***(1.58–3.26) |
| | Primary education | 2.65***(1.96–3.59) | 4.92***(4.01–6.03) | 6.82***(4.92–9.45) |
| | More than primary | 3.15*(1.18–8.42) | 6.05***(4.27–8.58) | 16.24***(7.93–33.29) |
| Literate | Illiterate | 1.36(0.93–2.01) | 2.35***(1.83–3.03) | 3.16***(1.75–5.70) |
| | Primary education | 2.85***(1.96–4.12) | 5.21***(4.23–6.41) | 7.55***(5.70–9.99) |
| | More than primary | 5.77***(2.86–11.67) | 5.87***(4.68–7.37) | 9.32***(7.31–11.87) |
| **Model 2: adjusted for gender and age** | | | | |
| Illiterate | Illiterate | 1.00 | 1.55***(1.27–1.88) | 1.40*(1.01–1.95) |
| | Primary education | 2.08***(1.54–2.80) | 2.66***(2.16–3.27) | 3.30***(2.37–4.58) |
| | More than primary | 2.31(0.90–5.90) | 3.13***(2.20–4.46) | 6.99***(3.63–13.43) |
| Literate | Illiterate | 1.47(1.00–2.15) | 1.90***(1.50–2.42) | 2.42***(1.44–4.04) |
| | Primary education | 2.26***(1.54–3.31) | 3.05***(2.48–3.75) | 4.03***(3.05–5.33) |
| | More than primary | 3.76**(1.75–8.07) | 3.34***(2.65–4.21) | 4.49***(3.53–5.72) |
| **Model 3: adjusted for gender, age and baseline health** | | | | |
| Illiterate | Illiterate | 1.00 | 1.12(0.93–1.36) | 1.02(0.67–1.57) |
| | Primary education | 1.23(0.93–1.62) | 1.40**(1.13–1.72) | 1.63**(1.19–2.23) |
| | More than primary | 1.90(0.84–4.32) | 1.63**(1.16–2.29) | 3.34***(1.67–6.64) |
| Literate | Illiterate | 1.08(0.73–1.59) | 1.14(0.90–1.46) | 1.59(0.87–2.93) |
| | Primary education | 1.52*(1.08–2.15) | 1.49***(1.22–1.84) | 1.83***(1.37–2.44) |
| | More than primary | 2.34(1.22–4.47) | 1.55***(1.23–1.96) | 2.08***(1.63–2.67) |

* p<0.05, ** p<0.01, *** p<0.001.

proportion of individuals lost to follow-up assigned to the unhealthy category increases, the results demonstrate greater stability, and the upper and lower bound values of the 95% confidence intervals show a gradual convergence towards the study's results. Notably, as the percentage of scenarios increases, a more pronounced gradient emerges, which supports the hypothesis. Health outcomes are better for participants with higher socioeconomic status accumulation across three life stages. The detailed table results are presented in S1 File.

## Discussion

An observed gradient in health outcomes corresponds to the accumulation of SES, and the results also highlight the independent effects of each SES component on health. The gradient improvement in health associated with SES is evident and has been consistently found to be related to a wide range of diseases and overall health [24,25]. These effects persist into old age, as demonstrated by previous research [26,27] and corroborated by the findings of this study.

Detailing further, the accumulation of SES sets the tone for baseline health and affects the rate of decline. From a life course perspective, the development of bodily functions is a dynamic process. The maturity plateau, shaped by early life risk factors throughout childhood, precedes a progressive decline in bodily functions with age, with the rate of decline influenced by adult exposures [28]. Based on the theory, our study focuses on health shaped by the accumulation of SES, which includes baseline health and the aging process during the survey years. Controlling for gender, baseline age, and baseline health in Model 3 allows us to observe the rate of decline, thereby ascertaining that the aging process is influenced by SES, which supports the hypothesis.

Additionally, the data indicate that the effects of each SES indicator are independent. Previous studies that classify those subgroups of populations with the same SES indicator as having uniform health conditions overlook the fact that their health status varies due to different SES levels experienced throughout their life trajectories. Each stage of life leaves its mark on the individual. Disadvantaged SES imprints on individuals, even if they later achieve better SES. For instance, those with the highest income but lower educational attainment exhibit comparably worse health, a status not offset by their later advantaged SES. As a result, it is essential to acknowledge that these subgroups are not regarded as having the same health status. Nonetheless, the health statuses of these subgroups are not purely independently different. Given that these subgroups share the same SES indicators with two other different SES, their health statuses are relevant to one another.

When examining health outcomes, it is crucial to recognize that the SES of the previous life stage not only correlates with later health but also perpetuates into later life. The established positive association between health and income in adulthood finds its roots in childhood [29]. Having a literate father significantly increases the likelihood of the son achieving higher education, and higher educational attainment often correlates with a better income. An observable tendency for individuals to inherit their SES from earlier life stages is consistent with prior research, indicating social reproduction. Parents' resources play a lasting influence on their children's academic success, even if these effects remain independent of children's own resources and diminish over time. Early and later socialization have independent and cumulative effects [30,31]. Therefore, this interaction of all three SES indicators cannot be overlooked when considering the association between SES and health.

Among all SES indicators, participants' educational attainment plays a pivotal role, especially in IADL disability. Previous studies suggest that education holds significant potential in providing years of independent life, reducing both the duration and proportion of dependent years [15,32,33]. Delving into the reasons, education not only represents SES but also encompasses additional aspects. It contributes to the enhancement of health literacy [34], the adoption of healthy behaviors, and access to healthcare [33].

The intricate nature of the SES-health relationship makes it difficult to fully disentangle all the interconnections. While two-layer gradient of health outcomes had been demonstrated in previous research [11,35], our study extends this trend to old age health with a three-layer gradient. Individuals' SES reinforces itself over time, shaping health accordingly [36]. Through the application of this study design, we have detected distinctions among these subgroups. The analysis indicates that each stage of life leaves an indelible mark on the individual. Each subsequent socioeconomic status exerts its own independent effect on future health, while the preceding status sets the baseline. The cumulative exposure to SES factors shapes health step by step.

## Limitations

This study's unit of analysis focuses on the individual. While factors at the community and neighborhood (meso) and national and international (macro) levels can also impact individuals' health [37], they were not considered in this analysis. For instance, the sample population was drawn from 331 Taiwanese townships, where living standards vary slightly. Consequently, household income may hold different meanings for different participants, a factor we did not control for. Additionally, government policies and the broader social environment may exert influence on individuals. The proliferation of higher education began in the 1960s in Western nations [38] and the 1990s in Taiwan. Most participants' fathers from that era had limited access to schooling. The younger generation has generally attained higher levels of education, in line with our study's observations. As literacy has improved over time, educational attainment may carry different implications for individuals of different ages [39], and even more so for their fathers.

Secondly, the cohort effect remains a limitation we could not fully address. The negative association between age and health can be elucidated by biological aging and cohort effects: health tends to decline with age, and younger cohorts generally exhibit better health than older cohorts [2,40]. Socioeconomic conditions for the current generation in Taiwan

have progressively improved over time [15]. Conversely, certain studies have discovered that the younger generation exhibits poorer health compared to the previous generation [41–44], for reasons that are not yet fully understood. Additionally, there is a greater heterogeneity in the health outcomes influenced by education among older age groups than among younger ones [2,45]. The health status across different generations varies due to changes in the social environment. Owing to the small sample size of this dataset, the generational issue was not addressed in this article. Nevertheless, the consistent phenomenon of health being shaped by SES endures. These findings may be generalizable to more advanced studies.

Limitations of the methodology in this study should also be noted. There are two potential sources of overestimation. The three health outcomes analyzed in this study are dichotomized, with one ADL indicator and each of the six IADL indicators categorized into two groups: "functionally independent" or "limited." Focusing on positive situations within the population, if any indicator displays "limited," we classified as "limited." This approach may result in a higher percentage of individuals in the unhealthy group than the actual count.

Secondly, we categorized individuals as "lost to follow-up" without distinguishing between those who went missing and those who passed away. All those categorized as "lost to follow-up" were included in the unhealthy group. Consequently, a higher percentage of unhealthy individuals than the actual count may be presented. Moreover, while we included all participants until the last wave of the survey, the results may not fully represent the actual situation of the participants still remaining. Previous research by House and colleagues has shown that health disparities tend to diverge throughout most of life stages but converge in the elderly [26,46]. Nevertheless, age-related patterns in health trajectories may vary across different dimensions of SES and health [47]. Upon conducting a sensitivity analysis, we observe similar outcomes. Regardless of the different proportions of those lost to follow-up classified as unhealthy, the health gradient caused by the accumulation of socioeconomic status remains stable, though the magnitude attenuates. Given that participants lost to follow-up in cohort studies are rarely lost randomly [48], studies have found that these individuals tend to belong to lower socioeconomic status groups and experience more health problems, regardless of whether these differences are statistically significant [49–52]. As a result, we can only assert that this study observes a trend indicating that health is associated with the accumulation of SES; however, the extent to which this association holds requires further investigation.

## Conclusion

Despite these potential sources of overestimation, our study demonstrates stable gradient health outcomes attributable to the accumulation of SES. Although each subsequent SES exerts its own independent effect on future health, the preceding SES sets the baseline. Moreover, populations with the same SES indicator should not be treated as having uniform health conditions. This assumption overlooks the variation in health status caused by different SES levels experienced over the course of their life trajectories. Policies are better implemented in early life stages to enhance later health and SES. Nevertheless, it is never too late, as each SES at different life stages has its own effect and cannot be offset.

## Supporting information

**S1 File. Sensitivity analyses.**
(DOCX)

## Author contributions

**Conceptualization:** Hsiao-Hsiao Tan, Tung-liang Chiang.

**Data curation:** Hsiao-Hsiao Tan, Yi-Chan Lee.

**Formal analysis:** Yi-Chan Lee.

**Methodology:** Hsiao-Hsiao Tan, Yi-Chan Lee, Tung-liang Chiang.

**Project administration:** Hsiao-Hsiao Tan.

**Supervision:** Ya-Mei Chen, Tung-liang Chiang.

**Writing – original draft:** Hsiao-Hsiao Tan.

**Writing – review & editing:** Hsiao-Hsiao Tan, Yi-Chan Lee, Tung-liang Chiang.

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
