## [Decision Letter · Decision Letter 0]

Dear Dr. Tan,

We look forward to receiving your revised manuscript.

Kind regards,

Muhammad Fawad, PhD

Academic Editor

PLOS ONE

Journal Requirements:

Additional Editor Comments:

I have some suggestions to further improve the manuscript.

1.The authors should explain how categorizing all participants who were "lost to follow-up" as part of the unhealthy group might affect the study's conclusions. Could this approach lead to an overestimation of the association between socioeconomic status and poor health

outcomes?

2.How reliable is it to use a father's education as the only indicator of childhood socioeconomic status? Are there other measures that could provide a more comprehensive understanding of childhood socioeconomic status?

3.How do the observed cohort effects influence the interpretation of the cumulative socioeconomic status on health outcomes? Additionally, how generalizable are these findings to other populations or younger cohorts?

4.The manuscript contains several grammatical mistakes, specifically on Page# 6, Line 89, Page# 7, Line 114, Page #10, Line 158, and Page# 17, Line 205. I highly recommend conducting a thorough proofread to correct these errors and improve its overall clarity and

readability.

Reviewers' comments:

Reviewer's Responses to Questions

**Comments to the Author**

1. Is the manuscript technically sound, and do the data support the conclusions?

Reviewer #1: Yes

Reviewer #2: Partly

2. Has the statistical analysis been performed appropriately and rigorously?

Reviewer #1: Yes

Reviewer #2: No

3. Have the authors made all data underlying the findings in their manuscript fully available?

Reviewer #1: Yes

Reviewer #2: Yes

4. Is the manuscript presented in an intelligible fashion and written in standard English?

Reviewer #1: No

Reviewer #2: Yes

Reviewer #1: This study examines the relationships between socioeconomic status (SES) at different life stages and health outcomes in old age, using data from the Taiwan Longitudinal Study on Aging (TLSA). The authors analyze three SES indicators across the life course: father's education (childhood SES), participant's education (early adulthood SES), and monthly household income (late adulthood SES). They assess three health outcomes: self-rated health, ADL disability (bathing only), and IADL disability. The study employs Generalized Estimating Equations (GEE) models to investigate longitudinal patterns and the cumulative effects of SES on health outcomes.

Some Suggestions:

1-Clarify the specific research questions or hypotheses in the introduction.

2-would it be possible to Consider exploring potential mediating factors between SES and health outcomes, such as health behaviors or access to healthcare?

3-The study does not address potential reverse causality between SES and health, which could be particularly relevant for late adulthood SES (income) and health outcomes.

Reviewer #2: This study explores the relationship between socioeconomic status (SES) and health among elderly Taiwanese individuals. The manuscript addresses a relevant topic. However, after a careful reading, I have identified some areas that could be improved to enhance the clarity and quality of the manuscript.

METHODS

The Methods section is well-structured but could benefit from additional clarity and detail, especially regarding variables and analytical procedures. Suggestions include:

1.1. Describe the study design.

1.2. Detail how the SES variables were categorized and justify these choices. Also, describe how adjusted variables were collected and categorized.

1.3. Explain in more detail the ‘hot-deck’ imputation method used for handling missing data. Cite a reference for this technique from the scientific literature as well.

1.4. Provide more details about the choice of Generalized Estimating Equations (GEE) models, specifying the type of regression used (Poisson, Logistic, Negative Binomial ...). Also, detail the level of aggregation, wicth probably is the year or the wave.

1.5. Provide a figure with the model of Analysis Planning for variables organization (DAG - Directed Acyclic Graph).

1.6. Provide more details about the sampling process and the sample size calculation performed.

RESULTS

The Results section does not present the findings clearly and can be improved with more detail and structure.

2.1. Specify the demographic characteristics of the participants in the different waves, as well as response rates and follow-up losses. For example: “Of the initial 4,049 participants, 60% were women, and the average age was 70 years at baseline. In 2003, 1,743 participants remained, with a response rate of approximately 90% in subsequent waves.”

2.2. Provide a flowchart of the sample for each wave.

2.3. Create a first table with the sample description stratified by waves (five waves) and include the education variable in the row, as well as the other variables in the study.

2.4. Tables are not self-explanatory. Reorganize the columns or rows, as it is currently not possible to verify the reference categories in the regressions models.

DISCUSSION

3.1. Avoid being too emphatic in stating that the observed gradients in outcomes are due to cumulative effects, as the study design does not permit this conclusion. Clarify whether this is a cross-sectional or longitudinal cohort study. If the study followed the same individuals from 1989 onwards, it is a cohort study. However, if new individuals were included in each wave, it is cross-sectional, and the waves are time series that can be aggregated using GEE, but these are not cumulative effects, even if parental data were used. Please clarify this point.

3.2. What are the other individual and contextual possible confounding factors not addressed in this study but that might affect the results?

3.3- Discuss the aging process in more detail.

3.4. Similarly to the beginning of the discussion, reconsider the conclusion as it is currently too emphatic and deterministic, which might be biased due to the lack of methodological clarifications still needed.

I believe that with the suggested improvements, the manuscript will make a significant contribution to the literature on the relationship between SES and health among the elderly. I am available for further reviews and hope that my suggestions will help strengthen the article.

Regards.

**Do you want your identity to be public for this peer review?** For information about this choice, including consent withdrawal, please see our Privacy Policy

Reviewer #1: **Yes: ** Elham Faghihzadeh

Reviewer #2: No

---

## [Author Response · Author response to Decision Letter 1]

4 Oct 2024

【Editor Comments】

1. The authors should explain how categorizing all participants who were "lost to follow-up" as part of the unhealthy group might affect the study's conclusions. Could this approach lead to an overestimation of the association between socioeconomic status and poor health outcomes?

Response:

Thank you for noting that the overestimation of unhealthy data might have resulted from classifying all “lost to follow-up” individuals as unhealthy. We conducted sensitivity analyses to investigate the health outcomes with varying proportions of “lost to follow-up” individuals classified as unhealthy. These analyses were included in the statistical analyses section of the methods on Page# 11, Line 181 of the revised manuscript, with results concluded on Page# 23, Line 259 and the table results detailed in S1 File.

In the sensitivity analyses, we randomly assigned 80% and 90% of individuals “lost to follow-up” as unhealthy, and also tested the outcome when all lost to follow-up individuals were classified as missing. Comparing the sample size of those lost to follow-up with mortality count data from the 2007 TLSA research report, the calculated mortality rate among individuals classified as “lost to follow-up” across five study waves ranged from 83.9% to 92.5%. The comparative data are presented in Table 1.

*Table 1 Mortality rates among participants classified as “lost to follow-up” across five waves

In addition, in cohort studies, participants lost to follow-up are rarely lost randomly (Kristman et al., 2004). Studies have found that individuals lost to follow-up tend to belong to lower socioeconomic status groups and experience more health problems, whether these differences are statistically significant or not (Hoeymans et al., 1998; Launer et al., 1994; Mihelic & Crimmins, 1997; Norris, 1985).

The sensitivity analyses yielded similar results to the study. Health outcomes were better for participants with higher socioeconomic status accumulation across three life stages. The magnitude of the results for the 80% and 90% scenarios was attenuated, yet the overall associations and patterns remained stable. The findings from all participants lost to follow-up, categorized as missing, demonstrate an even more pronounced effect on SES accumulation, although they exhibit significant fluctuations and instability. This limitation has been covered in the discussion section; please refer to Page# 28, Line 351 in the revised manuscript.

2. How reliable is it to use a father's education as the only indicator of childhood socioeconomic status? Are there other measures that could provide a more comprehensive understanding of childhood socioeconomic status?

Response:

We are grateful for your reminder to explore other indicators of childhood SES. Common indicators of childhood SES include household income, and the education and occupation of both parents (Bradley & Corwyn, 2002; Kaplan et al., 2001). In the TLSA, data on mothers' socioeconomic status (SES) and childhood household income were not collected, while information on fathers' education and occupation was. By using fathers’ occupation as a measure of childhood SES, we conducted the GEE analysis and found similar results. Please see the results in Table 4-6 at the end of this answer of this question.

Earlier studies often emphasize fathers’ SES, as women's access to education and formal employment has increased only in more recent decades, and childhood household income is prone to recall bias when participants are asked to report it. Therefore, we have selected the father’s education and occupation as the indicators representing childhood SES. Table 2 is the comparative table of sample numbers stratified by fathers’ education and occupation as indicators.

*Table 2 Comparative sample sizes stratified by fathers’ education and occupation

The data reveal that SES from a preceding stage has a considerable effect on SES in the next stage, with large fluctuations in SES trajectories being rare. Moreover, a higher number of individuals with the most advantaged adulthood SES had literate fathers compared to those with white-collar fathers. The correlation coefficients for each indicator are presented in Table 3. According to Table 3, fathers’ education exhibits a higher correlation with all indicators in comparison to fathers’ occupation. As education serves not only as an indicator of SES but also as a precursor to occupation and income (Liberatos et al., 1988), we used fathers’ education as the proxy for childhood SES instead of occupation.

*Table 3 Spearman correlation coefficients for SES indicators and baseline health outcomes

*Table 4 GEE analysis of self-rated health (SRH): odds ratios (95% confidence intervals) across three life stages’ SES strata

*Table 5 GEE analysis of ADL disability (bathing only): odds ratios (95% confidence intervals) across three life stages’ SES strata

*Table 6 GEE analysis of IADL disability: odds ratios (95% confidence intervals) across three life stages’ SES strata

3. How do the observed cohort effects influence the interpretation of the cumulative socioeconomic status on health outcomes? Additionally, how generalizable are these findings to other populations or younger cohorts?

Response:

Thank you for pointing this out. We conducted a GEE analysis by separating participants into two pseudo-cohort groups to observe cohort effects, and found consistent results. Greater accumulation of SES is often linked to superior health outcomes. Please see the results in Table 7-8.

Results for the younger pseudo-cohort remain similar and are reinforced. Conversely, the older pseudo-cohort model is quite unstable. Hence, we conducted another GEE analysis, where all individuals lost to follow-up were classified as missing, as shown in Table 9. The results were akin to those in Table 8. Additionally, Table 10 displays the distribution of the older pseudo-cohort stratified by three SES indicators. It is observed that the survival rate increases with higher SES accumulation. Nevertheless, as reported by the Ministry of the Interior, the average life expectancy in Taiwan was 73.4 years in 1989 and 77.0 years in 2003. Eighty percent of the participant population was lost to follow-up, which might explain the model’s instability.

While we recognize the importance of this consideration, the cohort effect is unavoidable and cannot be analyzed in this manuscript because the dataset lacks sufficient variability. Cumulative effects have been demonstrated in previous studies (Smith et al., 1997), and we observed similar findings in this study, where individuals with more advantaged SES tended to exhibit better health. Despite the fact that contemporary populations generally have higher levels of educational attainment, similar outcomes should be expected, provided that health continues to be shaped by SES. These findings may be generalizable to more advanced studies. This limitation has been addressed in the discussion section, please refer to Page# 27, Line 331 in the revised manuscript.

*Table 7 GEE analysis of SRH: odds ratios (95% confidence intervals) across three life stages’ SES strata (participants aged 60-69 years)

*Table 8 GEE analysis of SRH: odds ratios (95% confidence intervals) across three life stages’ SES strata (participants aged 70-96 years)

*Table 9 GEE analysis of SRH: odds ratios (95% confidence intervals) across three life stages’ SES strata (participants aged 70-96 years and all lost to follow-up individuals are excluded)

*Table 10 Population size of the older pseudo-cohort stratified by three SES indicators

4. The manuscript contains several grammatical mistakes, specifically on Page# 6, Line 89, Page# 7, Line 114, Page #10, Line 158, and Page# 17, Line 205. I highly recommend conducting a thorough proofread to correct these errors and improve its overall clarity and readability.

Response:

Thank you for your suggestions. We have carefully revised the manuscript to enhance overall clarity and readability, and to correct grammatical errors. Please find the revised manuscript enclosed.

Here are the corrections for the specific points you mentioned:

*Page# 6, Line 89 > Page# 6, Line 103

Origin: “Out of the initial 4,049 participants, only 1,743 remained in the study during the 2003 survey, with 2,306 lost to follow-up.”

Correct: “Of the initial 4,049 participants, 1,743 were retained in the study for the 2003 survey, while 2,306 were lost to follow-up.”

*Page# 7, Line 114 > Page# 8, Line 137

Origin: “Monthly household income levels of respondents were classified into three categories: low, medium, and high.”

Correct: “Respondents' monthly household income levels were categorized into three strata: low (< 3,000 NTD), medium (3,000-19,999 NTD), and high (≥ 20,000 NTD).”

*Page #10, Line 158 > Page# 12, Line 199

Origin: “Generally, populations with more advantageous SES characteristics tend to exhibit better overall health, even after adjusting for demographic factors and baseline health.”

Correct: “Generally, populations characterized by more advantageous SES tend to exhibit better overall health outcomes, even after controlling for demographic factors and baseline health.”

*Page# 17, Line 205 > Page# 22, Line 236

Origin: “Still, superior early and late adulthood SES significantly reflects a lesser decline in health.”

Correct: “Yet, the rate of decline is influenced by SES accumulation, with increased SES accumulation significantly associated with a lesser decline in health.”

【Reviewer #1】

1- Clarify the specific research questions or hypotheses in the introduction.

Response:

Thank you for pointing this out. The research hypotheses presented in the introduction have been revised as follow. Please also refer to Page# 5, Line 83 in the revised manuscript.

“The purpose of this study is to investigate how health trajectory in old age is shaped by the accumulation of socioeconomic status across three life stages.”

2- would it be possible to Consider exploring potential mediating factors between SES and health outcomes, such as health behaviors or access to healthcare?

Response:

Thank you for highlighting this. While we agree that this is an important consideration, it is beyond the scope of this manuscript. This study examines the impact of SES on health, considering that the small size of subgroup samples could limit further analysis of mediating effects. The correlation coefficients for SES indicators, health behaviors, and baseline health outcomes are presented in Table 11. Table 11 reveals that health behaviors exhibit a mild correlation with all other indicators, which may pose challenges for the analysis of mediating effects.

*Table 11 Spearman correlation coefficients for SES indicators, health behaviors, and

baseline health outcomes

As the mediating effect of health behaviors and psychosocial characteristics is well-documented (Lynch et al., 1997; Macintyre, 1997), our study is specifically aimed at examining the direct relationship between SES and health.

3- The study does not address potential reverse causality between SES and health, which could be particularly relevant for late adulthood SES (income) and health outcomes.

Response:

Thank you for highlighting this. While we agree that this is an important consideration, it is beyond the scope of this manuscript due to the need for advanced analytical methods or the inclusion of additional variables.

The effect of health on SES is acknowledged but was not included in our study. The issue of bidirectional causality has been acknowledged since the 1980s, as noted in the Black Report. Table 12 displays the correlation coefficients for SES indicators and baseline health outcomes. While late adulthood SES is significantly related to three health indicators, the effects are weaker than those from early adulthood SES (education), as education represents more than just an SES indicator.

Moreover, most prior research has concentrated on how SES affects health rather than the reverse, as the impact of health on SES is relatively minor compared to the impact of SES on health. We used an early-life health indicator as an example to test the correlation and found similar results. An indicator of early life health is included in Table 12. The health indicator at age 16 was measured in 2003, as it was not surveyed in 1989. Consequently, those lost to follow-up are all classified as unhealthy, potentially leading to an overestimation of early-life unhealthiness. Table 12 reveals the minor yet significant impact of early-life health on SES and health in later life. Our research is limited to examining the effect of SES on health, whilst acknowledging that early-life health can have a minor but profound impact on later-life SES and health (Blane et al., 1993; Haas, 2008; Haas, 2006; West, 1991).

The intricate nature of the SES-health relationship makes it difficult to fully disentangle all the interconnections. Our study focuses on examining the direct relationship between SES and health.

*Table 12 Spearman correlation coefficients for SES indicators and health outcomes

【Reviewer #2】

METHODS

The Methods section is well-structured but could benefit from additional clarity and detail, especially regarding variables and analytical procedures. Suggestions include:

1.1. Describe the study design.

Response:

Thank you for your suggestions. The study design has been added to the revised manuscript; please see Page# 7, Line 117.

“This study aims to explore how cumulative exposure to socioeconomic status (SES) at different life stages shapes health trajectories in later life. SES data were collected in the baseline year, and health trajectories were analyzed using Generalized Estimating Equations (GEE). This design allowed for the examination of baseline health and subsequent changes over time.”

1.2. Detail how the SES variables were categorized and justify these choices. Also, describe how adjusted variables were collected and categorized.

Response:

The three chosen SES indicators are as follows. This study utilized SES indicators across three life stages: childhood, early adulthood, and late adulthood. These indicators included the educational attainment of participants' fathers, the participants' own educational attainment, and monthly household income. The revised version is included on Page# 8, Line 137 of the updated manuscript.

Regarding father’s education, most participants’ fathers from that era had limited access to schooling. This provided an opportunity to examine the health effects of parental education on their offspring, specifically whether the parent received education or not. In the dataset, approximately half of the fathers were literate, while the other half were not.

Educational opportunities improved over time, allowing us to divide participants’ education levels into three categories to capture health disparities between the more advantaged and disadvantaged groups.

A similar rationale was applied to income categorization to enhance the visibility of health differences. The 1989 survey collected data on participants’ monthly household income, categorized as: (1) less than 3,000 NTD, (2) 3,000-4,999 NTD, (3) 5,000-9,999 NTD, (4) 10,000-14,999 NTD, (5) 15,000-19,999 NTD, (6) 20,000-49,999 NTD, and (7) above 50,000 NTD. According to Taiwan’s Ministry of Health and Welfare, the minimum living wage in 1991 was 3,000 NTD. The Directorate-General of Budget, Accounting and Statistics, Executive Yuan, reported an average salary of 21,247 NTD in 1989. Therefore, we used 3,000 NTD and 20,000 NTD as cut points to divide participants into three income groups: low (< 3,000 NTD), medium (3,000-19,999 NTD), and high (≥ 20,000 NTD).

Recognizing the importance of sociodemographic characteristics in determining health, gender and age were included as covariates. The age range of participants was 60 to 96 years. They were classified into five groups, each with a five-year interval starting at age 60, with the last group including those aged 80 and above. This study aims to examine how the accumulation of SES across three life stages informs both baseline health and subsequent health changes. Therefore, baseline health was also controlled for.

---

## [Decision Letter · Decision Letter 1]

Dear Dr. Tan,

Thank you for submitting your manuscript to PLOS ONE. After careful consideration, we feel that it has merit but does not fully meet PLOS ONE’s publication criteria as it currently stands. Therefore, we invite you to submit a revised version of the manuscript that addresses the points raised during the review process.

We look forward to receiving your revised manuscript.

Kind regards,

Muhammad Fawad, PhD

Academic Editor

PLOS ONE

Journal Requirements:

Reviewers' comments:

Reviewer's Responses to Questions

**Comments to the Author**

Reviewer #3: (No Response)

2. Is the manuscript technically sound, and do the data support the conclusions?

Reviewer #3: Yes

3. Has the statistical analysis been performed appropriately and rigorously?

Reviewer #3: Yes

4. Have the authors made all data underlying the findings in their manuscript fully available?

Reviewer #3: (No Response)

5. Is the manuscript presented in an intelligible fashion and written in standard English?

Reviewer #3: (No Response)

Reviewer #3: Editor Comments

1. Categorizing "lost to follow-up" participants as unhealthy

• The authors should extend their sensitivity analysis to include a broader range of scenarios (e.g., 30%, 50%, 70%). This would better demonstrate the robustness of their conclusions across varying assumptions about the health status of those lost to follow-up.

• The authors could conduct an analysis comparing the SES variables of those lost to follow-up with those who remained in the study. This is an effective way to provide empirical support for the assumption that individuals lost to follow-up tend to have worse SES.

2. Cohort effects and generalizability of findings

• Response: The authors addressed cohort effects through analyses of pseudo-cohorts and discussed the limitations of dataset variability.

• Recommend that the authors include age as a time-varying covariate in their Generalized Estimating Equations (GEE) model. This would allow them to better differentiate between the effects of biological aging and cohort effects.

Reviewer #2’s Comment

1- Study design and methodological clarifications

It could be improved with more detail and clarity about the study’s design and methodology such as whether this is a cohort study or explain key design elements, such as the timeline, follow-up intervals and a brief description of how participants were recruited and followed over time.

2- Detail about SES variables:

Why did not authors use methods like Principal Component Analysis (PCA) to extract a single composite SES variable?

3-Results

I could not find any changes regarding comment 2.1 in the manuscript, or I may be missing something.

**Do you want your identity to be public for this peer review?** For information about this choice, including consent withdrawal, please see our Privacy Policy

Reviewer #3: No

---

## [Author Response · Author response to Decision Letter 2]

7 Mar 2025

Our heartfelt thanks go to the editor and reviewers. Your thoughtful questions have inspired us to delve deeper into our analysis and improve our explanations.

---

## [Decision Letter · Decision Letter 2]

Dear Dr. Tan,

We look forward to receiving your revised manuscript.

Kind regards,

Sreeram V. Ramagopalan

Academic Editor

PLOS ONE

Journal Requirements:

Reviewers' comments:

Reviewer's Responses to Questions

**Comments to the Author**

Reviewer #3: (No Response)

Reviewer #4: All comments have been addressed

2. Is the manuscript technically sound, and do the data support the conclusions?

Reviewer #3: Yes

Reviewer #4: Yes

3. Has the statistical analysis been performed appropriately and rigorously?

Reviewer #3: Yes

Reviewer #4: Yes

4. Have the authors made all data underlying the findings in their manuscript fully available?

Reviewer #3: (No Response)

Reviewer #4: Yes

5. Is the manuscript presented in an intelligible fashion and written in standard English?

Reviewer #3: (No Response)

Reviewer #4: Yes

Reviewer #3: (No Response)

Reviewer #4: Does the study design sufficiently account for potential biases introduced by the loss of follow-up participants? It may benefit from a more direct explanation of how these objectives specifically relate to the study's aims regarding socioeconomic status (SES) and health trajectories.

The exclusion of mountainous areas merits more justification. How might this exclusion impact the generalizability of the findings?

Lines 127-129: Please provide more details regarding the reasons for participant attrition. Were there common factors among those lost to follow-up that could introduce selection bias?

Lines 160-167: Consider adding context about the significance of the chosen income categories. How do these thresholds correlate with the economic context of Taiwan during the relevant years?

Lines 172-185: While the selection of health indicators appears rational, the focus on bathing as an Activities of Daily Living (ADL) measure seems somewhat narrow. Please justify the choice of bathing over other potential indicators, such as eating or dressing.

The abbreviations for TLSA, SES, and others were not introduced in Figure 1 and Tables. Please clarify these terms for better understanding.

Regarding the Self-Rated Health (SRH) question with a 5-point Likert scale, which specific points were considered indicative of "good" health?

**Do you want your identity to be public for this peer review?** For information about this choice, including consent withdrawal, please see our Privacy Policy

Reviewer #3: **Yes: ** Somayeh Momenyan

Reviewer #4: **Yes: ** Farzane Ahmadi

---

## [Author Response · Author response to Decision Letter 3]

10 Jun 2025

Does the study design sufficiently account for potential biases introduced by the loss of follow-up participants? It may benefit from a more direct explanation of how these objectives specifically relate to the study's aims regarding socioeconomic status (SES) and health trajectories.

Response:

We appreciate your reminder. The constraints inherent in this study restricted the ability to achieve a fully robust design. Given that the majority of participants lost to follow-up were deceased, they were categorized as unhealthy to better analyze the association between socioeconomic status (SES) and health trajectories. Comprehensive sensitivity analyses were conducted to ensure the stability and reliability of the data.

Comparing the sample size of those lost to follow-up with mortality count data from the 2007 TLSA research report, the calculated mortality rate among individuals classified as “lost to follow-up” across five study waves ranged from 83.9% to 92.5%. The comparative data are provided in Table 1 for further reference.

Table 1 Mortality rates among participants classified as “lost to follow-up” across

five waves

The exclusion of mountainous areas merits more justification. How might this exclusion impact the generalizability of the findings?

Response:

We appreciate your reminder. The relationship between SES and health is firmly established. Moreover, the multifaceted interactions both within various SES dimensions and between SES and health exhibit considerable complexity. This study aimed to further explore this relationship by elucidating the connections between SES at three distinct life stages and later health trajectories. The health gradients were examined from multiple perspectives. An interesting observation emerged, which may merit further exploration. These findings may be generalizable to more advanced studies.

Lines 127-129: Please provide more details regarding the reasons for participant attrition. Were there common factors among those lost to follow-up that could introduce selection bias?

Response:

Thank you for pointing this out. As shown in Table 1, the majority of participants lost to follow-up were deceased, while the remaining individuals either declined participation, were repeatedly untraceable, or lost contact entirely. Nonetheless, in each wave, only approximately 10% of the remaining participants were not included in the investigation. This survey was conducted under the coordination of the Health Promotion Administration, Ministry of Health and Welfare, and benefited significantly from local authorities' assistance. The reliability of the data was ensured despite certain losses.

However, in cohort studies, participants lost to follow-up are rarely lost randomly (Kristman et al., 2004). Studies have found that individuals lost to follow-up tend to belong to lower socioeconomic status groups and experience more health problems, whether these differences are statistically significant or not (Hoeymans et al., 1998; Launer et al., 1994; Mihelic & Crimmins, 1997; Norris, 1985).

The remaining participants generally have higher SES, potentially leading to selection bias. While we acknowledge the significance of this factor, the loss of participants was inevitable. Despite the fact, similar outcomes should be expected, provided that health continues to be shaped by SES.

Lines 160-167: Consider adding context about the significance of the chosen income categories. How do these thresholds correlate with the economic context of Taiwan during the relevant years?

Response:

The 1989 survey collected data on participants’ monthly household income, categorized as: (1) less than 3,000 NTD, (2) 3,000-4,999 NTD, (3) 5,000-9,999 NTD, (4) 10,000-14,999 NTD, (5) 15,000-19,999 NTD, (6) 20,000-49,999 NTD, and (7) above 50,000 NTD. According to Taiwan’s Ministry of Health and Welfare, the minimum living wage in 1991 was 3,000 NTD. The Directorate-General of Budget, Accounting and Statistics, Executive Yuan, reported an average salary of 21,247 NTD in 1989. Therefore, we used 3,000 NTD and 20,000 NTD as cut points to divide participants into three income groups: low (< 3,000 NTD), medium (3,000-19,999 NTD), and high (≥ 20,000 NTD).

Lines 172-185: While the selection of health indicators appears rational, the focus on bathing as an Activities of Daily Living (ADL) measure seems somewhat narrow. Please justify the choice of bathing over other potential indicators, such as eating or dressing.

Response:

The ADLs include the following: “(1) Ambulating: The Extent of an individual's ability to move from 1 position to another and walk independently; (2) Feeding: The Ability of an individual to feed oneself; (3) Dressing: The ability to select appropriate clothes and to put them on; (4) Personal hygiene: The ability to bathe and groom oneself and maintain dental hygiene, nail, and hair care; (5) Continence: The ability to control bladder and bowel function; (6) Toileting: The ability to get to and from the toilet, use it appropriately, and clean oneself afterward” (Edemekong et al., 2025).

In the 1989 interview, participants were asked about their ability to: (1) independently purchase daily necessities, such as soap, toothpaste, and medicine; (2) manage financial matters, including paying bills and handling change; (3) make phone calls; (4) maintain personal hygiene, specifically through bathing; (5) climb stairs; (6) walk a distance of 200–300 meters; (7) perform labor-intensive tasks, such as floor cleaning and agricultural work; (8) manage transportation, either by driving or arranging alternative modes; (9) lift objects weighing approximately 12 kg; (10) perform squatting movements; (11) raise arms above head level; (12) manipulate objects using fine finger movements; and (13) maintaining a standing position for a continuous duration of two hours.

Due to the incomplete ADL data in the 1989 interview, only bathing was selected as the representative ADL indicator for this study. Nevertheless, in Taiwan, difficulty bathing is the most frequently cited limitation in ADL. Having persistent difficulties bathing is an independent risk of long-term nursing home admission (Gill et al., 2006).

The abbreviations for TLSA, SES, and others were not introduced in Figure 1 and Tables. Please clarify these terms for better understanding.

Response:

We appreciate your reminder. Kindly review the revised Figure 1, the updated Table 1, and the supporting information.

Regarding the Self-Rated Health (SRH) question with a 5-point Likert scale, which specific points were considered indicative of "good" health?

Response:

Thank you for pointing this out. For the purposes of this study, we designated "very good" and "good" health as "good," whereas the other three classifications were categorized as "poor" health, given our analytical focus on overall health status.

---

## [Editor Report · Decision Letter 3]

Socioeconomic inequalities in health among elderly people in Taiwan: a life course perspective

PONE-D-24-26589R3

Dear Dr. Tan,

We’re pleased to inform you that your manuscript has been judged scientifically suitable for publication and will be formally accepted for publication once it meets all outstanding technical requirements.

Kind regards,

Sreeram V. Ramagopalan

Academic Editor

PLOS ONE
---

## [Editor Report · Acceptance letter]

PONE-D-24-26589R3

PLOS ONE

Dear Dr. Tan,

I'm pleased to inform you that your manuscript has been deemed suitable for publication in PLOS ONE. Congratulations! Your manuscript is now being handed over to our production team.

Kind regards,

on behalf of

Dr. Sreeram V. Ramagopalan

Academic Editor

PLOS ONE